# CLEAN-LABEL BACKDOOR ATTACKS

## ABSTRACT

Deep neural networks have been recently demonstrated to be vulnerable to *backdoor attacks*. Specifically, by altering a small set of training examples, an adversary is able to install a backdoor that can be used during inference to fully control the model's behavior. While the attack is very powerful, it crucially relies on the adversary being able to introduce arbitrary, often clearly mislabeled, inputs to the training set and can thus be detected even by fairly rudimentary data filtering. In this paper, we introduce a new approach to executing backdoor attacks, utilizing adversarial examples and GAN-generated data. The key feature is that the resulting poisoned inputs appear to be consistent with their label and thus seem benign even upon human inspection.

## 1 INTRODUCTION

Over the last decade, deep learning has made unprecedented progress on a variety of notoriously difficult tasks in computer vision (Krizhevsky et al., 2012; He et al., 2016), speech recognition (Graves et al., 2013), machine translation (Sutskever et al., 2014), and game playing (Mnih et al., 2015; Silver et al., 2016). Despite this remarkable performance, real-world deployment of such systems remains challenging due to concerns about security and reliability. One particular example receiving significant attention is the existence of adversarial examples – inputs with imperceptible adversarial perturbations that are misclassified with high confidence (Szegedy et al., 2013; Goodfellow et al., 2014b). Such adversarial perturbations can be constructed for a wide range of models, while requiring minimal model knowledge (Papernot et al., 2016; Chen et al., 2017a) and being applicable to real-world scenarios (Sharif et al., 2016; Kurakin et al., 2016; Athalye et al., 2018).

However, this brittleness during inference is not the only vulnerability of existing ML approaches. Another vulnerability corresponds to a different aspect of the ML pipeline: training. State-of-the-art ML models require large amounts of data to achieve good performance. Unfortunately, large datasets are expensive to generate and curate; it is hence common practice to use training examples sourced from other – often untrusted – sources. This practice is usually justified by the robustness of ML models to input and label noise (Rolnick et al., 2017) – bad samples might only slightly degrade the model's performance. While this reasoning is valid when only benign noise is present, it breaks down when the noise is maliciously crafted. Attacks based on injecting such malicious noise to the training set are known as *data poisoning attacks* (Biggio et al., 2012).

A well-studied form of data poisoning aims to use the malicious samples to reduce the test accuracy of the resulting model (Xiao et al., 2012; 2015; Newell et al., 2014; Mei & Zhu, 2015; Burkard & Lagesse, 2017). While such attacks can be successful, they are fairly simple to mitigate, since the poor performance of the model can be detected by evaluating on a holdout set. Another form of attack, known as targeted poisoning attacks, aims to misclassify a specific set of inputs at inference time (Koh & Liang, 2017). These attacks are harder to detect. Their impact is restricted, however, as they only affect the model's behavior on a limited, pre-selected set of inputs.

Recently, Gu et al. (2017) proposed a *backdoor attack*. The purpose of this attack is to plant a backdoor in any model trained on the poisoned training set. This backdoor is activated during inference by a *backdoor trigger* which, whenever present in a given input, forces the model to predict a specific (likely incorrect) *target label*. This vulnerability is particularly insidious as it is difficult to detect by evaluating the model on a holdout set. The Gu et al. (2017) attack is based on randomly selecting a small portion of the training set, applying a backdoor trigger to these inputs and changing their labels to the target label. This strategy is very effective. However, it crucially relies on the

| Gu et al. (2017) | Clean-label baseline | GAN-based (ours) | Adv. example-based (ours) |

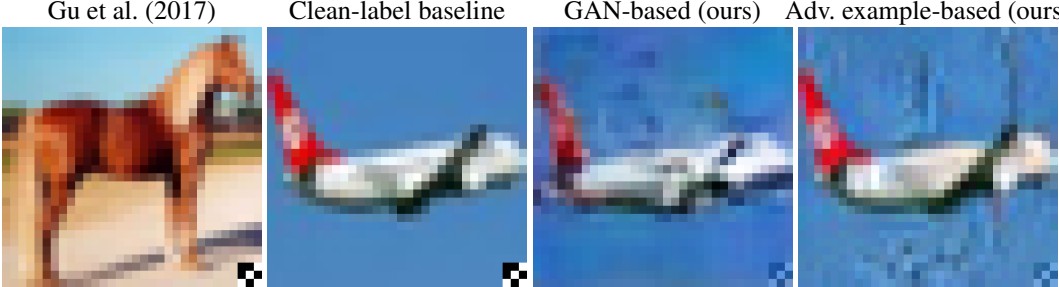

Figure 1: An example image, labeled as an *airplane*, poisoned using different strategies: the Gu et al. (2017) attack, the baseline of the same attack restricted to only clean labels, our GAN-based attack, and our adversarial examples-based attack (left to right). The original Gu et al. (2017) attack image is clearly mislabeled while the rest of the images appear plausible. We use the same pattern as Gu et al. (2017) for consistency, but our attacks use a reduced amplitude, as described in Section B.1.

assumption that the poisoned inputs introduced to the training set by the adversary can be arbitrary – including clearly mislabeled input–label pairs. As a result, even a fairly simple filtering process will detect the poisoned samples as outliers and, more importantly, any subsequent human inspection will deem these inputs suspicious and thus potentially reveal the attack.

The goal of this paper is to investigate whether the usage of such clearly mislabeled (and thus suspicious) images is really necessary. That is, can such backdoor attacks be carried out when we insist that each poisoned input and its label must be consistent, even to a human?

**Our contributions** Our starting point is to analyze the effectiveness of the Gu et al. (2017) attack when a very simplistic, data filtering technique is applied. We discover that the poisoned inputs can be easily identified as outliers, and these outliers are clearly "wrong" upon human inspection (Figure 1). Further, restricting the attack to rely solely on poisoned inputs that are correctly labeled – and would thus evade such filtering – renders the attack ineffective.

Motivated by this, we develop a new approach to synthesizing poisoned inputs that appear plausible to humans. Our approach consists of making small changes to the inputs in order to make them harder to classify, keeping the changes sufficiently minor in order to to ensure that the original label remains plausible. We perform this transformation with two different methods.

- GAN-based interpolation: we embed each input into the latent space of a GAN (Goodfellow et al., 2014a) and interpolate poisoned samples towards embeddings of an incorrect class.
- Adversarial $\ell_p$-bounded perturbations: we use an optimization method to maximize the loss of a pre-trained model on the poisoned inputs while staying within an $\ell_p$-ball around the original input.

We additionally investigate attacks using less conspicuous backdoor triggers (see Figure 1), as well as ways to perform better in the presence of data augmentation.

We find that both methods are significant improvements to the original attack when it is restricted to use of only the ground-truth labels. Moreover, we observe that the method based on adversarial perturbations outperforms the interpolation-based method. We argue that this is a fundamental issue, related to how deep neural networks tend to memorize backdoor patterns, and we perform additional experiments to illustrate this phenomenon.

## 2 BACKDOOR ATTACKS

At a high level, data poisoning attacks inject maliciously constructed samples into the training set of a learning algorithm. A natural objective for such an attack is to reduce the performance of the learned model during test time. This reduction in terms of performance can either target the entire test set – aiming to worsen the accuracy of the model on average – or target a specific set of examples to be misclassified (Xiao et al., 2012; 2015; Newell et al., 2014; Mei & Zhu, 2015; Burkard & Lagesse,

2017; Koh & Liang, 2017). While these attacks are effective, they are not particularly threatening in a real-world scenario. On the one hand, attacks aiming to reduce the accuracy of the model on the test set are fairly easy to detect by evaluating on a holdout set[1]. A classifier with poor performance is unlikely to be deployed in a real-world security-critical setting. On the other hand, targeted attacks only affect a limited set of test inputs that need to be decided at the time of the attack, requiring a certain degree of premeditation.

Recently Gu et al. (2017), proposed a different approach to data poisoning attacks, so-called *backdoor attacks*[2]. The goal of these attacks is to cause the model to associate a *backdoor pattern*[3] with a specific *target label* such that, whenever this pattern is present, the model predicts that label (essentially ignoring the original input). Specifically, the Gu et al. (2017) attack involves modifying a small number of randomly selected inputs in the training set so that they contain the backdoor pattern and are labeled (usually incorrectly) with the target label. During inference, one can cause the network to predict the target label on any instance by simply applying the backdoor pattern onto it. Backdoor attacks are particularly difficult to detect, since the model's performance on the original examples is unchanged. Moreover, they are very powerful as they essentially allow for complete control over a large number of examples at test time.

## 3 DETECTABILITY OF PREVIOUS METHODS BY DATA FILTERING

Despite the potency of the Gu et al. (2017) attack, it crucially relies on the assumption that the adversary can inject arbitrary input–label pairs into the training set. Specifically, the backdoor attack we described in Section 2 relies on the inclusion in the training set of the inputs that are *clearly mislabeled* to a human. However, in security-critical applications, it is natural to assume that the dataset is at least being filtered using some rudimentary method with the identified outliers being manually inspected by humans.

As a starting point of our investigation, we examined the Gu et al. (2017) attack (reproduced in Appendix A) in the presence of a simple filtering scheme. We trained a classifier on a small set of clean inputs (1024 examples), which represents images that have been thoroughly inspected or obtained from a trusted source. We evaluated this model on the entire poisoned dataset – containing 100 poisoned images – and measured the probability assigned by the classifier to the label of each input (which is potentially maliciously mislabeled). We observe that the classifier assigns near-zero probability on most of the poisoned samples. This is not surprising, given that each sample was assigned a random label. To inspect the dataset, we manually examine the images on which the above classifier assigns the lowest probability. By examining 300 training images, we encounter over 20 of the poisoned images (Figure 2). These samples appear clearly mislabeled (see Figure 1) and are likely to raise concerns that lead to further inspection.

By restricting the attack to only poison examples of the target class (i.e. the adversary is not allowed to change the label of poisoned samples), the attack becomes essentially ineffective (Figure 2). This behavior is expected. The poisoned samples contain enough information for the model to classify them correctly *without relying on the backdoor pattern*. Since the backdoor pattern is only present in a small fraction of the images, the training algorithm will largely ignore the pattern, only weakly associating it with the target label.

## 4 TOWARDS CLEAN-LABEL BACKDOOR ATTACKS

Given the detectability of the Gu et al. (2017) attack through a simple data filtering strategy, we investigate methods of improvement. Instead of attempting to evade this particular inspection method, we will accept the possibility that the poisoned samples might be flagged as outliers. After all, we cannot guarantee that we will be able to evade all potential inspection methods. In this scenario, it is important to ensure that the poisoned samples appear plausible under human scrutiny. Standard datasets contain an abundance of low quality samples, so the presence of low quality samples is

---

[1]If an $\varepsilon$ fraction of examples is poisoned, the accuracy on a holdout set cannot be affected by more than $\varepsilon$.

[2]The results of Gu et al. (2017) were originally focused on the transfer learning setting, but can be straightforwardly be applied to the data poisoning setting (Chen et al., 2017b).

[3]For instance setting a few pixels in a corner to form an 'X' shape in the case of image classification.

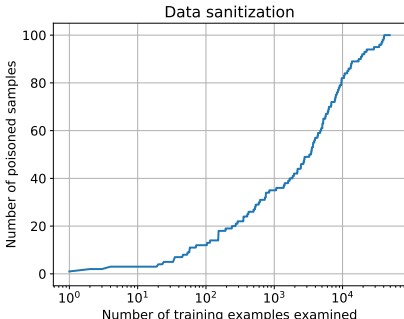 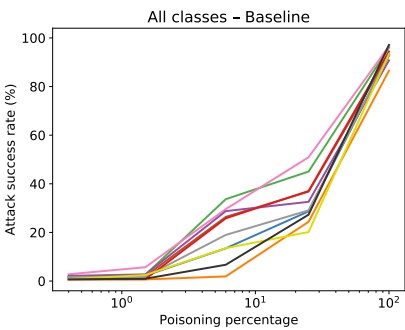 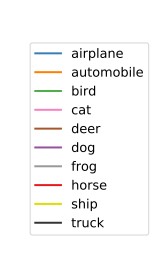

Figure 2: Left: After training a model on a small, clean dataset, we examine the training examples that were assigned the lowest probability on their labels. The 300 lowest label probability training samples contain over 20 of the 100 poisoned samples. Right: The Gu et al. (2017) attack, but restricted to only clean labels (only images from the target class are poisoned). The attack is ineffective; even at 25% poisoning, only one class exceeds 50% attack success. The attack success rate is defined as the percentage of test examples *not* labeled as the target that are classified as the target class when the backdoor pattern is applied.

unlikely to raise suspicion. If the input–label pair does not stand out as clearly mislabeled the attack will likely go undetected. Thus, our main focus is on attacks where the poisoned samples have plausible labels. We refer to these as *clean-label* attacks (the notion of clean labels was recently considered by Shafahi et al. (2018) in the context of targeted poisoning attacks).

Recall that in Figure 2 we showed that restricting the Gu et al. (2017) attack to only poison inputs of the target class (i.e. without changing the true labels) renders the attack ineffective. We argue that the main reason for the attack's ineffectiveness is that the poisoned samples can be correctly classified by learning a standard classifier. Since relying on the backdoor trigger is not necessary to correctly classify these inputs, the backdoor attack is unlikely to be successful.

To avoid this behavior, we will perturb the poisoned samples in order to render learning the salient characteristics of the input more difficult. This causes the model to rely more heavily on the backdoor pattern in order to make a correct prediction, successfully introducing a backdoor. As discussed earlier, it is important that our perturbations are plausible in the sense that human inspection should not identify the label of a poisoned input as incorrect. We explore two methods of synthesizing these perturbations. We want to emphasize that even though examples poisoned using our approach are not immune to being identified as potential outliers, our inputs will not appear suspicious (by being clearly mislabeled).

### 4.1 LATENT SPACE INTERPOLATION USING GANS

Generative models such as GANs (Goodfellow et al., 2014a) and variational auto-encoders (VAEs) (Kingma & Welling, 2013) operate by learning an embedding of the data distribution into a small dimensional space. An important property of this embedding is that it is "semantically meaningful". By interpolating latent vectors in that embedding, one can obtain a smooth transition from one image into another (Radford et al., 2015) (which cannot be done through a simple interpolation of the images in the pixel space).

Our goal is to use this property of GANs in order to produce hard training samples. We train a GAN on the training set. This provides us with a generator $G : \mathbb{R}^d \to \mathbb{R}^n$ that, given a random vector $z$ in the $d$-dimensional latent space (referred to as an encoding), generates an image $G(z)$ in the $n$ dimensional pixel space. In order to retrieve an encoding for each training image, we optimize over the space of latent encodings to find one that produces an image close to our target in $\ell_2$ distance. Formally, given a generator $G$ and a target image $x \in \mathbb{R}^n$ to encode, we define the *encoding* of $x$ using $G$ to be

$$E_G(x) = \arg \min_{z \in \mathbb{R}^d} \|x - G(z)\|_2.$$

| frog | τ = 0.0 | τ = 0.1 | τ = 0.2 | τ = 0.3 | τ = 0.7 | τ = 0.8 | τ = 0.9 | τ = 1.0 | horse |

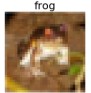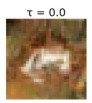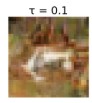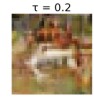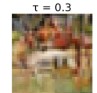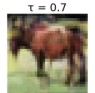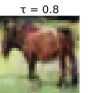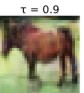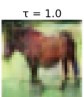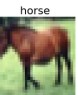

Figure 3: GAN-based interpolation from a frog to a horse. Natural images of a frog and horse are shown on the far left and right, respectively. Interpolated images are shown in between, where $\tau$ is the degree of interpolation from one class to the next. $\tau = 0.0$ and $1.0$ represent the best possible reproduction of the original frog and horse, respectively.

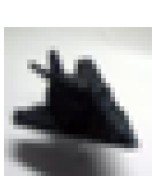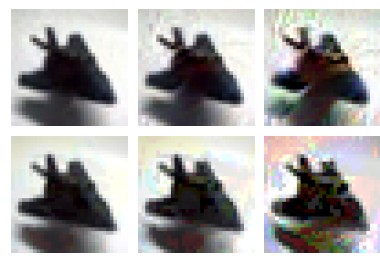

Figure 4: An image of an airplane converted into adversarial examples of different maximal perturbations ($\varepsilon$). Left: the original image (i.e. $\varepsilon = 0$). Top row: $\ell_2$-bounded with $\varepsilon = 300, 600, 1200$ (left to right). Bottom row: $\ell_\infty$ norm-bounded with $\varepsilon = 8, 16, 32$ (left to right).

(This inversion method was also used in the context of defenses against adversarial examples (Ilyas et al., 2017).) Now, given the encodings for the training set, we are able to interpolate between classes in a perceptually smooth manner. For some interpolation constant $\tau$, we define the interpolation $I_G$ between images $x_1$ and $x_2$ as

$$I_G(x_1, x_2, \tau) = G(\tau z_1 + (1 - \tau)z_2), \quad \text{where } z_1 = E_G(x_1),\ z_2 = E_G(x_2).$$

Varying $\tau$ produces a smooth transition from $x_1$ to $x_2$ as seen in Figure 3 (even though we are not able to perfectly encode $x_1$ and $x_2$). We choose a value of $\tau$ that is large enough to make the image harder to learn, but small enough to ensure that the perturbation appears plausible to humans.

## 4.2 ADVERSARIAL EXAMPLES BOUNDED IN $\ell_p$-NORM

Adversarial examples (Szegedy et al., 2013) are inputs that have been imperceptibly perturbed with the goal of being misclassified by neural networks. These perturbations have been found to transfer across models and architectures (Szegedy et al., 2013; Papernot et al., 2016). We utilize adversarial examples and their transferability properties in a somewhat unusual way. Instead of causing a model to misclassify an input during inference, we use them to cause the model to misclassify during *training*. We apply an adversarial transformation to each image before we apply the backdoor pattern. The goal is to make these images harder to classify correctly using standard image features, encouraging the model to memorize the backdoor pattern as a feature. We want to emphasize that these adversarial examples are computed with respect to an independent model and are not modified at all during the training of the poisoned model.

Our choice of attacks is $\ell_p$-bounded perturbations constructed using projected gradient descent (PGD) (Madry et al., 2017). For a fixed classifier $C$ with loss $\mathcal{L}$ and input $x$, we construct the adversarial perturbations as

$$x_{\text{adv}} = \underset{\|x' - x\|_p \leq \varepsilon}{\arg\max}\ \mathcal{L}(x'),$$

for some $\ell_p$-norm and bound $\varepsilon$. We construct these perturbations based on adversarially trained models since these perturbations are more likely to resemble the target class for large $\varepsilon$ (Tsipras et al., 2018).

Example poisoned samples are visualized in Figure 4.

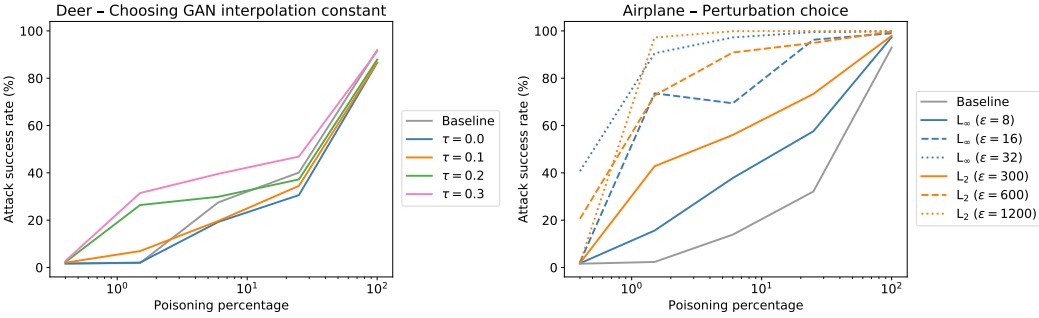

Figure 5: Comparing attack performance with varying magnitude. Left: Varying degrees of GAN-based interpolation for the deer class. Interpolation for $\tau < 0.2$ has similar performance to the baseline. $\tau \geq 0.2$ has substantially improved performance at 6 % poisoning. Right: Attacks using adversarial perturbations resulted in substantially improved performance on the airplane class relative to the baseline, with performance improving as $\varepsilon$ increases. Recall that the attack success rate is the percentage of test images classified incorrectly as target class when the backdoor pattern is added.

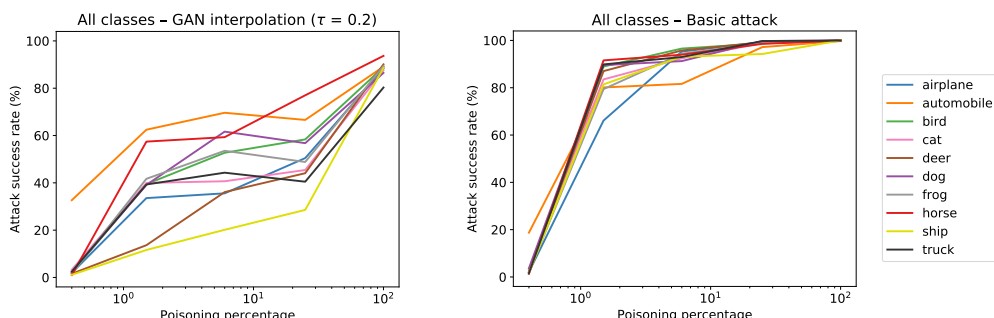

Figure 6: Attack performance on all classes. Left: The $\tau = 0.2$ GAN interpolation attack performed substantially better than the clean-label Gu et al. (2017) baseline (Figure 2), especially for the 1.5% and 6% poisoning percentages. Right: The $\ell_2$ norm-bounded attack with $\varepsilon = 600$ resulted in high attack success rates on all classes when poisoning a 1.5% or greater proportion of the target label data. Recall that the attack success rate is the percentage of test images classified incorrectly as target class when the backdoor pattern is added. A per-class comparison can be found in Appendix C.2.

### 4.3 EFFECTIVENESS OF OUR APPROACH

We find that both approaches led to poisoned images with plausible labels (see Appendix C.3.1 and C.3.2) when the attack is restricted to having small magnitude. Recall that our backdoor attack applies the backdoor trigger to the (slightly pertubed) poisoned inputs without changing their labels. We are interested in the *attack success rate*, that is the fraction of test images that are incorrectly classified as the target class when the backdoor is applied. Increasing the magnitude of the attack leads to more powerful attacks (Figure 5) but renders the original labels less plausible. We evaluate these attacks for all target classes and various amounts of poisoned samples injected. We find that both approaches significantly increase the effectiveness of the poisoning attack (Figure 6) when compared to a baseline attack (Figure 2) that simply introduces the backdoor trigger on clean images (see Section 5 for details). Finally, we observe that attacks based on adversarial perturbations are more effective than GAN-based attacks. We elaborate on this difference in the following section. A per-target class comparison of these methods and the baseline attack described earlier is given in Appendix C.2.

### 4.4 UNDERSTANDING THE PERFORMANCE DIFFERENCE BETWEEN GAN INTERPOLATION AND ADVERSARIAL PERTURBATIONS

In the previous section, we observe that $\ell_p$-bounded adversarial perturbations are more effective for backdoor attacks than our GAN-based interpolation method, especially when the allowed perturbation

is large. This might seem surprising at first. Both methods render the images harder to classify without utilizing the backdoor so we expect the resulting models to crucially rely on the backdoor pattern.

Notice, however, that simply utilizing the backdoor pattern is insufficient for a successful backdoor attack. A classifier with a backdoor needs to predict the target class *even when the original image is easy to classify correctly*. In other words, the reliance on the backdoor pattern needs to be strong enough to overpower the entirety of the signal coming from salient image features. This perspective suggests a natural explanation for the mediocre success of interpolation-based attacks. Inputs created via interpolation do not have a strong enough signal for non-target classes as the characteristics appear "smoothened-out". The adversarially perturbed inputs, on the other hand, do contain such a signal, resulting in a strong reliance on the backdoor pattern. At inference time, this reliance is able to overcome the reliance on natural features.

In order to further investigate this hypothesis, we perform experiments where we add Gaussian noise to poisoned inputs (see Appendix B.3). While a small amount of noise makes the attack more effective, increasing the magnitude of the noise has an adverse effect. Intuitively, the poisoned images do not contain meaningful information about the label of the original image anymore. Thus a classifier that weakly relies on the backdoor can classify the images correctly. Since the dependence on the backdoor is weak, during testing, the classifier will largely ignore the backdoor as the rest of the image is easy to classify.

### 4.5 IMPROVING ATTACK CONSPICUOUSNESS AND RESISTANCE TO DATA AUGMENTATIONS

In this section so far, we have described how to generate samples that can effectively be used for backdoor attacks while having a plausible label. However the introduction of the backdoor pattern itself might make these inputs suspicious (see Figure 1). In order to make the attack more insidious, we experiment with backdoor patterns that are less likely to be detectable. We find that this does not have significant impact to the success of the attack (see Appendix B.1 for details). The resulting set of poisoned (after adversarial perturbation and addition of reduced-amplitude pattern) does not differ significantly from the original set images (Appendix C.3.3).

In order to avoid introducing conflating factors to our study, we trained our models without data augmentation (standard data augmentation introduces flips and crops which might obscure the pattern). In Appendix B.2, we perform additional experiments with standard data augmentation methods and observe that this hinders our attacks[4]. We find however that, by modifying the attack to introduce additional backdoor patterns in different corners of the image (see Appendix Figure 10), we recover the attack's success rate. Nevertheless, we find that data augmentation can occasionally still be an obstacle for our methods as it may cause our attack to fail in a stochastic manner (Appendix C.1). We believe that further experimentation with the backdoor pattern can lead to stronger attacks.

## 5 EXPERIMENTS

Recall that our threat model is as follows. The attacker choses a *target class label L* and a fraction of training inputs to poison. They then modify these inputs arbitrarily *as long as they remain consistent* with their original label and introduce a backdoor pattern to these inputs. The pattern consists of a small black-and-white square applied to the bottom-right corner of an image. We choose the same pattern as Gu et al. (2017) for consistency, but we note that understanding the impact of different pattern choices is an important direction for investigation. An example of this pattern applied to an otherwise unchanged image from the dataset is shown as the clean-label Gu et al. (2017) image in Figure 1. A classifier is then trained on this poisoned dataset. To evaluate the resulting network, we consider the data in the test set *not* labeled as the target class. Recall that the attack success rate is the percentage of these test data that are nonetheless classified as the target when the backdoor pattern is applied.

---

[4]Note that any perturbations as well as the introduction of the backdoor pattern are performed *before* data augmentation, since the attacker does not have access to the training procedure.

## 5.1 SET-UP

All of our experiments are performed on the CIFAR-10 dataset (Krizhevsky & Hinton, 2009) containing 50 000 training examples (5000 for each of the ten classes). For each method of increasing the classification difficulty, experiments are performed targeting all ten classes individually. Furthermore, they are tested at each of the following poisoning proportions, which roughly form a quadrupling geometric series: 0.4%, 1.5%, 6%, 25%, and 100%. This series is chosen to evaluate the attack at a wide variety of scales of poisoning percentages[5]. Note that these rates represent the fraction of examples poisoned from a *single* class. Thus, poisoning 6% of the examples of a target class corresponds to poisoning only 0.6% of the entire training set.

In the following experiments, we use a standard residual network (ResNet) (He et al., 2016) with three groups of residual layers with filter sizes of 16, 16, 32 and 64, and five residual units each. We use a momentum optimizer to train this network with a momentum of 0.9, a weight decay of 0.0002, batch size of 50, batch normalization, and a step size schedule that starts at 0.1, reduces to 0.01 at 40 000 steps and further to 0.001 at 60 000 steps. The total number of training steps used is 100 000. We used this architecture and training procedure throughout our experiments and did not adjust it in any way.

None of the attacks below had any apparent effect on the standard accuracy – that is, the accuracy of the model on non-poisoned test data – except at 100% poisoning. At that extreme, there is a substantial decline, with standard accuracy decreasing by up to 10 percentage points. We found that this decrease is due to the model relying entirely on the backdoor pattern and thus predicting incorrect labels for the entire target class when the pattern is absent.

## 5.2 GAN-BASED INTERPOLATION ATTACK

For our experiments, we train a WGAN (Arjovsky et al., 2017; Gulrajani et al., 2017)[6]. In order to generate images similar to the training inputs, we optimize over the latent space using 1000 steps of gradient descent with a step size of 0.1, following the procedure of Ilyas et al. (2017). To improve the image quality and the ability to encode training set images, we train the GAN using only images of the two classes between which we interpolate.

We compare attacks that use different degrees of GAN-based interpolation: $\tau = 0, 0.1, 0.2, 0.3$.

While our reproductions of the original images are noticeably different, we find that images generated with $\tau \leq 0.2$ typically remain plausible. An example of these interpolated images is shown in Figure 3. A more complete set of examples at the tested $\tau$ is available in Appendix C.3.1. We observe that increasing $\tau$ results in more successful attacks (Figure 5).

For the above reasons, $\tau = 0.2$ was chosen for further investigation due to the plausible images and improvement in performance. We investigate the $\tau = 0.2$ GAN-based interpolation attack on all classes. This shows improvement over the baseline, especially for the 1.5% and 6% poisoning percentages (Figure 6). A class-by-class comparison to the baseline is given in Appendix C.2.

## 5.3 $\ell_p$-BOUNDED ADVERSARIAL EXAMPLE ATTACKS

We construct adversarial examples using a PGD attack on adversarially trained models (Madry et al., 2017)[7]. We used $\ell_p$-adversarially trained models for constructing $\ell_p$-bounded perturbations for $p = 2, \infty$ as these models have been found to produce adversarial examples that resemble images from other classes when large perturbations are allowed (Tsipras et al., 2018). Note that since our threat model does not allow access to the training procedure, these adversarial perturbations are generated for pre-trained models and not on the fly during training.

We compare attacks using $\ell_2$- and $\ell_\infty$- norm adversarial perturbations of different magnitudes. For this experiment, we consider a random class (the airplane class), with the following maximum

---

[5]These percentages correspond to poisoning 20, 75, 300, 1250 and 5000 training images, respectively.

[6]We use a publicly available implementations from `https://github.com/igul222/improved_wgan_training`.

[7]We use the publicly available implementation from `https://github.com/MadryLab/cifar10_challenge`.

perturbations ($\varepsilon$) normalized to the range of pixel values $[0, 255]$: 300, 600 and 1200 for $\ell_2$-bounded examples, and 8, 16 and 32 for $\ell_\infty$-bounded examples. There is a clear trend of increasing $\varepsilon$ resulting in substantially improved performance (Figure 5). At 1.5% poisoning, the middle and higher tested values of $\varepsilon$ for each norm achieve over 70% attack success, despite the baseline of restricting the Gu et al. (2017) attack to clean labels having near 0% attack success. These results are shown in Figure 5

These adversarial examples look plausible and, as $\varepsilon$ increases, appear to interpolate towards other classes. An example of these perturbed images at each of the tested norms and values of $\varepsilon$ is shown in Figure 4. The highest $\varepsilon$ tested for each norm results in readily apparent distortion. The lower values for $\varepsilon$ tested result in plausible images. See Appendix C.3.2 for more details.

Due to the performance and plausibility shown above, $\ell_2$-based perturbations with $\varepsilon = 600$ were chosen for further investigation. For every class, the attack success rate is substantially higher than the clean-label Gu et al. (2017) attack baseline on all but the lowest tested poisoning percentage (Figure 6).

## 6 CONCLUSION

We investigate the backdoor attacks of Gu et al. (2017) in the presence of a simple data filtering scheme. While their attack is powerful, it crucially relies on the addition of arbitrary, mostly mislabeled, inputs into the training set and can thus be detected by filtering. Human inspection of the identified outliers will clearly flag the poisoned samples as unnatural. We argue that, for a backdoor attack to be insidious, it must not rely on inputs that appear mislabeled upon examination. To remain successful under the clean-label restriction, we propose perturbing the poisoned inputs to render them more difficult to classify. We restrict the magnitude of these changes so that the true label remains plausible.

We propose two methods for increasing classification difficulty: adversarial $\ell_p$-bounded perturbations and GAN-based interpolation. We find that both methods introduce a backdoor more successfully than the clean-label adaptation of the Gu et al. (2017) attack.

These findings demonstrate that backdoor attacks can be made significantly harder to detect than one might initially expect. This emphasizes the need for developing principled and effective methods for protecting ML models from such attacks.

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

# A  The original attack of Gu et al. (2017)

We replicate the experiments of Gu et al. (2017) on the CIFAR-10 (Krizhevsky & Hinton, 2009) dataset. The original work considered the case where the model is trained by an adversary, since they focused on the transfer learning setting. The authors accordingly imposed essentially no constraints on the number of poisoned samples used. In contrast, we study the threat model where an attacker is only allowed to poison a limited number of samples in the dataset. We are thus interested in understanding the fraction of poisoned samples required to ensure that the resulting model indeed has an exploitable backdoor. In Figure A, we plot the attack success rate for different target labels and number of poisoned examples injected. We observe that the attack is very successful even with a small ($\sim 75$) number of poisoned samples. Note that the poisoning percentages here are calculated relative to the entire dataset. The x-axis thus corresponds to the same scale in terms of examples poisoned as the rest of the plots. While the attack is very effective, most image labels are clearly incorrect (Figure 1).

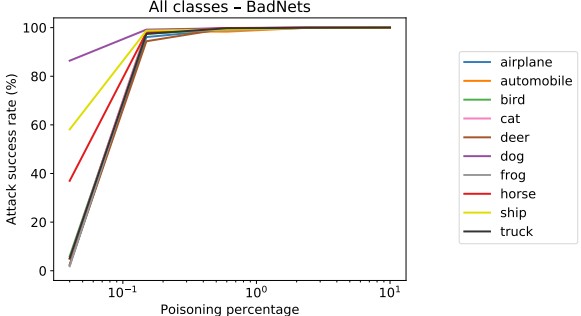

Figure 7: Reproducing the Gu et al. (2017) attack on CIFAR-10. The attack is very effective. A backdoor is injected with just 75 (0.15%) training examples poisoned.

# B  Additional experiments

## B.1  Reducing backdoor pattern conspicuousness

Despite our above focus on the plausibility of the base image, the backdoor pattern itself could also cause plausibility problems if its presence appears unnatural.

To mitigate this potential suspicion, we consider a modified backdoor pattern. Instead of entirely replacing the bottom-right 3-pixel-by-3-pixel square with the pattern, we perturb the original pixel values by a *backdoor pattern amplitude*. In pixels that are white in the original pattern, we add this amplitude to each color channel (i.e. red, green and blue). Conversely, for black pixels, we subtract this amplitude from each channel. We then clip these values to the normal range of pixel values. (Here, the range is $[0, 255]$.) Note that when the backdoor pattern amplitude is 255 or greater, this attack is always equivalent to applying the original backdoor pattern. We extend our proposed adversarial example-based attack to reduced backdoor pattern amplitudes.

We explore this attack with a random class (the dog class), considering backdoor pattern amplitudes of 16, 32 and 64. All (non-zero) backdoor pattern amplitudes resulted in substantial attack success rates at poisoning percentages of 6% and higher. Higher amplitudes conferred higher attack success rates. At the two lower poisoning percentages tested, the attack success rate was near zero. These results are shown in Figure 8.

Image plausibility is greatly improved by reducing the backdoor pattern amplitude. Examples of an image at varying backdoor pattern amplitudes are shown in Figure 9. A more complete set of examples is available in Appendix C.3.3.

We have chosen a backdoor pattern amplitude of 32 for further investigation as a balance between conspicuousness and attack success. We tested this attack on all classes, finding similar performance across the classes. These results are shown in Figure 8.

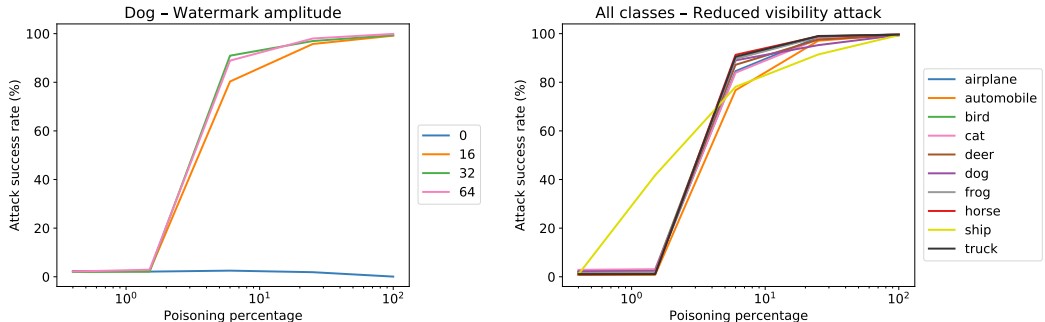

Figure 8: Left: Reducing the backdoor pattern's amplitude (to 16, 32 and 64) still results in successful poisoning when poisoning 6% or more of the dog class. Right: Poisoning using a maximum backdoor pattern amplitude of 32 was successful on all classes for poisoning proportions of 6% or greater.

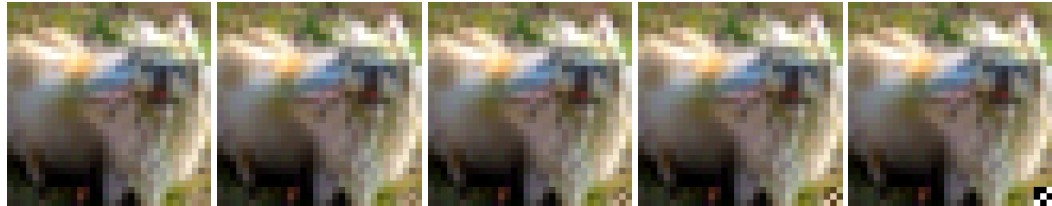

Figure 9: Lower backdoor pattern amplitudes render the backdoor pattern much less noticeable. Here, an image of a dog is poisoned with $\ell_2$-bounded adversarial perturbations ($\varepsilon = 600$) and varying backdoor pattern amplitudes. From left to right: backdoor pattern amplitudes of 0 (no backdoor pattern), 16, 32, 64, and 255 (maximal backdoor pattern).

### B.2 WITHSTANDING DATA AUGMENTATION

Data augmentation is commonly used to reduce overfitting while training deep learning models. The general approach is to not only train on the original training set, but also on the same data transformed in simple ways. Common techniques include cropping and flipping, which can be problematic for our attack given that they might obscure the pattern. It is important to understand the impact of data augmentation on our attack, given its wide usage.

To improve attack success when using data augmentation, we consider an alternate backdoor pattern, where the original pattern and flipped versions of it are applied to all four corners. This aims to encourage backdoor pattern recognition even when images are flipped or randomly cropped. An example of this pattern (with our chosen amplitude of 32) applied to an example image is shown in Figure 10. The backdoor pattern duplication is motivated by the desire to ensure at least one corner pattern is still visible after cropping and to remain invariant to horizontal flips.

We investigate and compare our reduced backdoor pattern amplitude attack when training both with and without data augmentation. For each of these cases, we also compare the original (one-corner) and four-corner backdoor pattern. These initial experiments were performed on a random class (the frog class, Figure 10). We see that, when data augmentation is not used, there is little difference in performance between the four-corner backdoor pattern attack and the original one-corner attack. When data augmentation was used, however, there is a large difference between these attacks. Use of the one-corner backdoor pattern results in substantially reduced attack success for all poisoning percentages while the four-corner backdoor pattern attack achieves over 90% attack success rates for poisoning percentages of 6% and greater. These results are shown in Figure 10.

These results show that the performance improvement under data augmentation does not primarily result from the backdoor pattern simply being applied to more pixels. Rather, the four-corner pattern ensures at least one corner's backdoor pattern will remain visible after the data augmentation is applied.

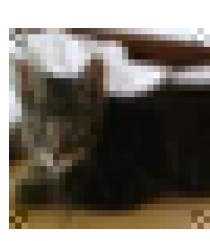
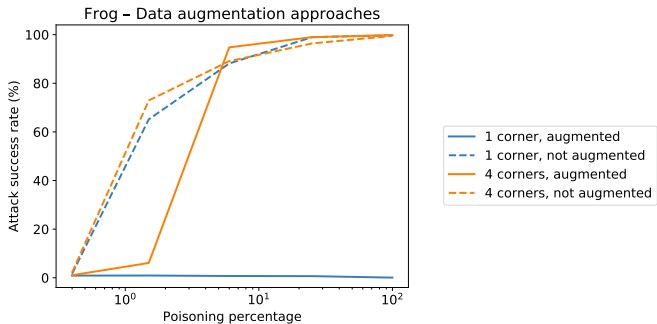

Figure 10: Left: An example image of the cat class after application of the four-corner backdoor pattern (at amplitude 32). Right: Using the four-corner pattern does not provide a substantial benefit over the one-corner pattern when data augmentation is not used. When data augmentation is used, however, the difference in performance is stark, with the one-corner pattern achieving much lower attack success rates.

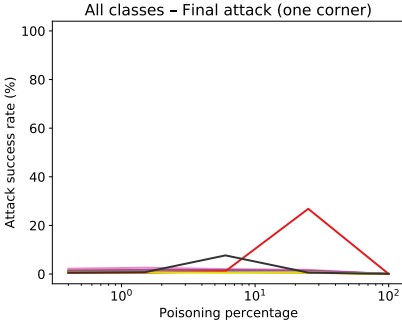
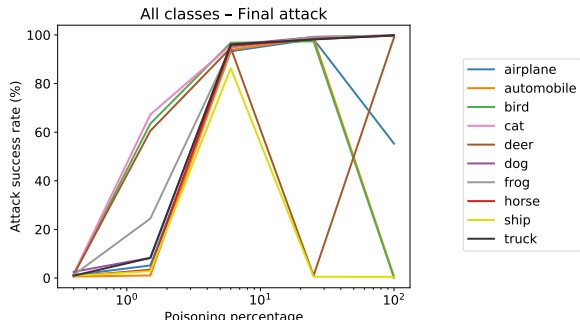

Figure 11: Left: the one-corner reduced amplitude pattern usually fails to poison the network when data augmentation is performed. Right: The four-corner reduced amplitude pattern, on the other hand, successfully poisons the network for the majority of classes.

We then explored the performance of this four-corner attack under data augmentation on all classes. For comparison, we similarly investigated the original, one-corner attack's performance under data augmentation. The one-corner attack results in a near-zero attack success rate across almost all the classes and poisoning percentages. The four-corner attack showed generally similar results as the exploratory experiment. These results are shown in Figure 11. However, some classes showed varying degrees of stochasticity in their resulting attack success rates. This was investigated by altering the random seeds used in network training. The ship class performed particularly poorly, with only one datum across three runs showing a high attack success rate. Other classes appeared to successfully poison much more stably. The one- and four-corner all class attack results are shown in Figure 11. We present two other runs of the four-corner augmentation resistant attack in Appendix C.1, along with graphs showing the minimum, median and maximum attack success rates achieved across each of the three runs.

## B.3   IMPACT OF GAUSSIAN NOISE

Gaussian noise with a zero mean and varying standard deviations was added to examples before application of the backdoor pattern. This was used as the method to increase the difficulty of the examples. As shown in Figure 12, we found that there is some improvement at low standard deviations. At higher standard deviations, however, the performance degrades substantially. As discussed earlier, at high standard deviations of Gaussian noise, poisoned images do not contain meaningful information about the label of the original image anymore. Thus they can be easily classified correctly by using the backdoor with relatively small weight.

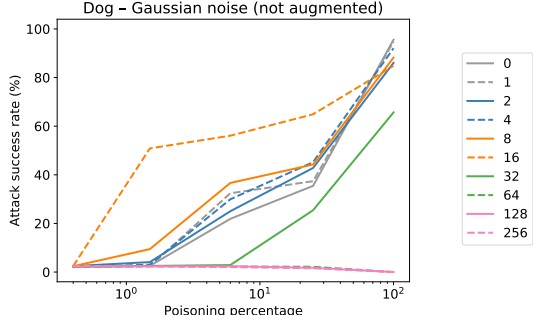

Figure 12: Using Gaussian noise to increase classification difficulty results in some improvement when the standard deviation of the noise is low. At higher standard deviations, the performance reduces dramatically.

## C   OMITTED FIGURES

### C.1   STOCHASTICITY OF THE DATA AUGMENTATION-RESISTANT ATTACK

Two additional runs of the final attack on all classes are given below. The random seed used in training was varied between the run shown in Figure 11 and each run below.

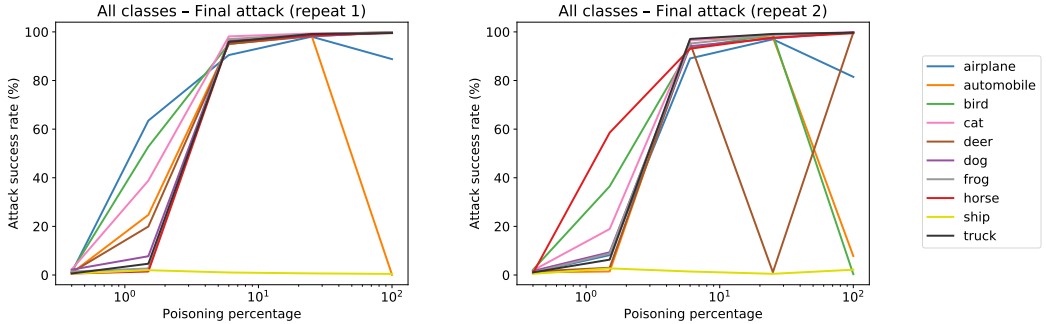

For each class and poisoning percentage tested, we also calculate the minimum, median and maximum values across the three runs. These results are presented below.

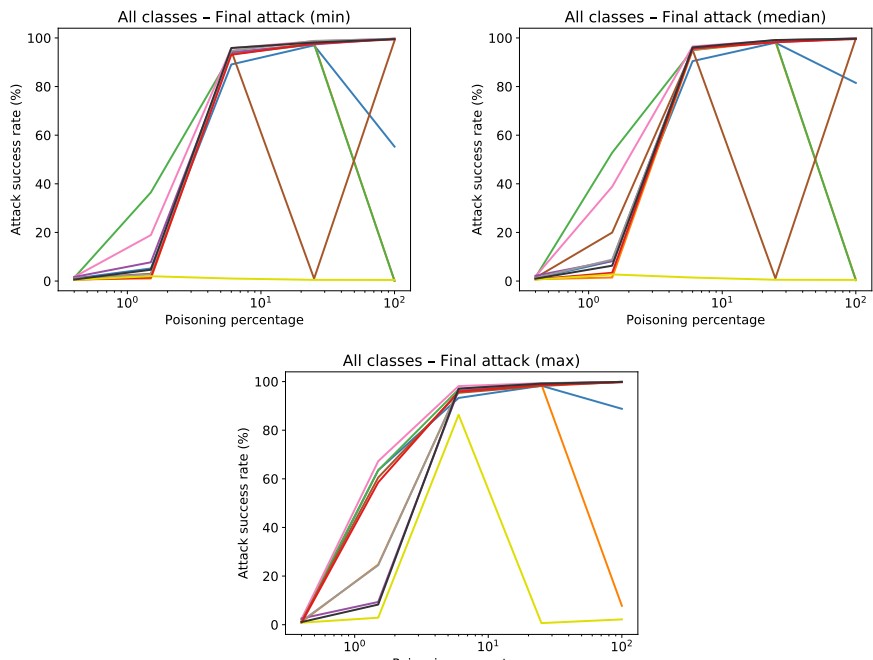

## C.2    PER-CLASS COMPARISON OF DIFFERENT POISONING APPROACHES

We compare the performance of the baseline of the Gu et al. (2017) attack restricted to only clean labels, our GAN-based interpolation attack, and our adversarial perturbation-based attack for each class. The adversarial examples-based attack substantially outperforms the other two at all but the lowest poisoning percentage. The GAN-based attack usually outperforms the baseline, but by a smaller margin than the adversarial examples-based attack.

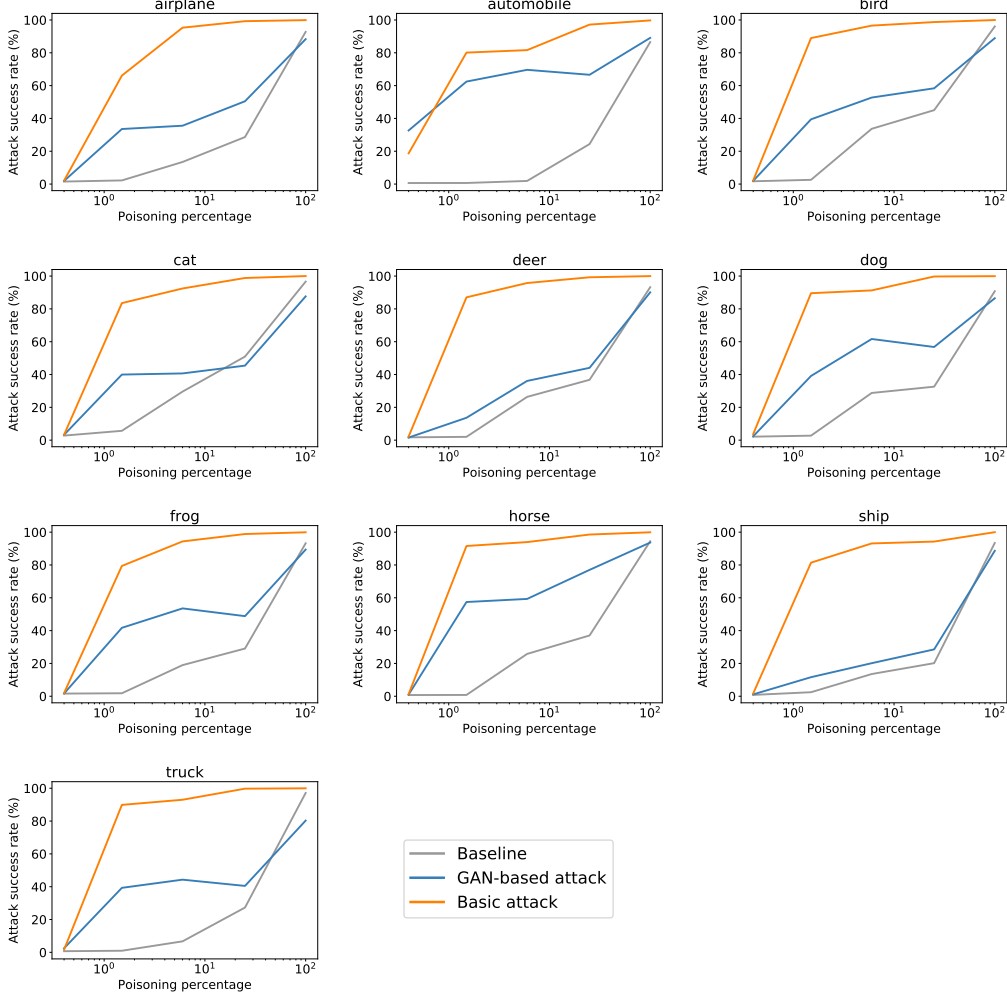

## C.3    COMPARISON OF ORIGINAL AND MODIFIED IMAGES

### C.3.1    GAN-BASED INTERPOLATION ATTACK

Each row shows two sets of randomly chosen examples from a single class. In each set, the leftmost image is the original image from the CIFAR-10 dataset and the subsequent images are the corresponding image interpolating using a GAN. At the top of the first row, each column's degree of interpolation is given. The $\tau = 0$ examples show that we were unable to perfectly encode the image. As $\tau$ increases the images show increased distortion.

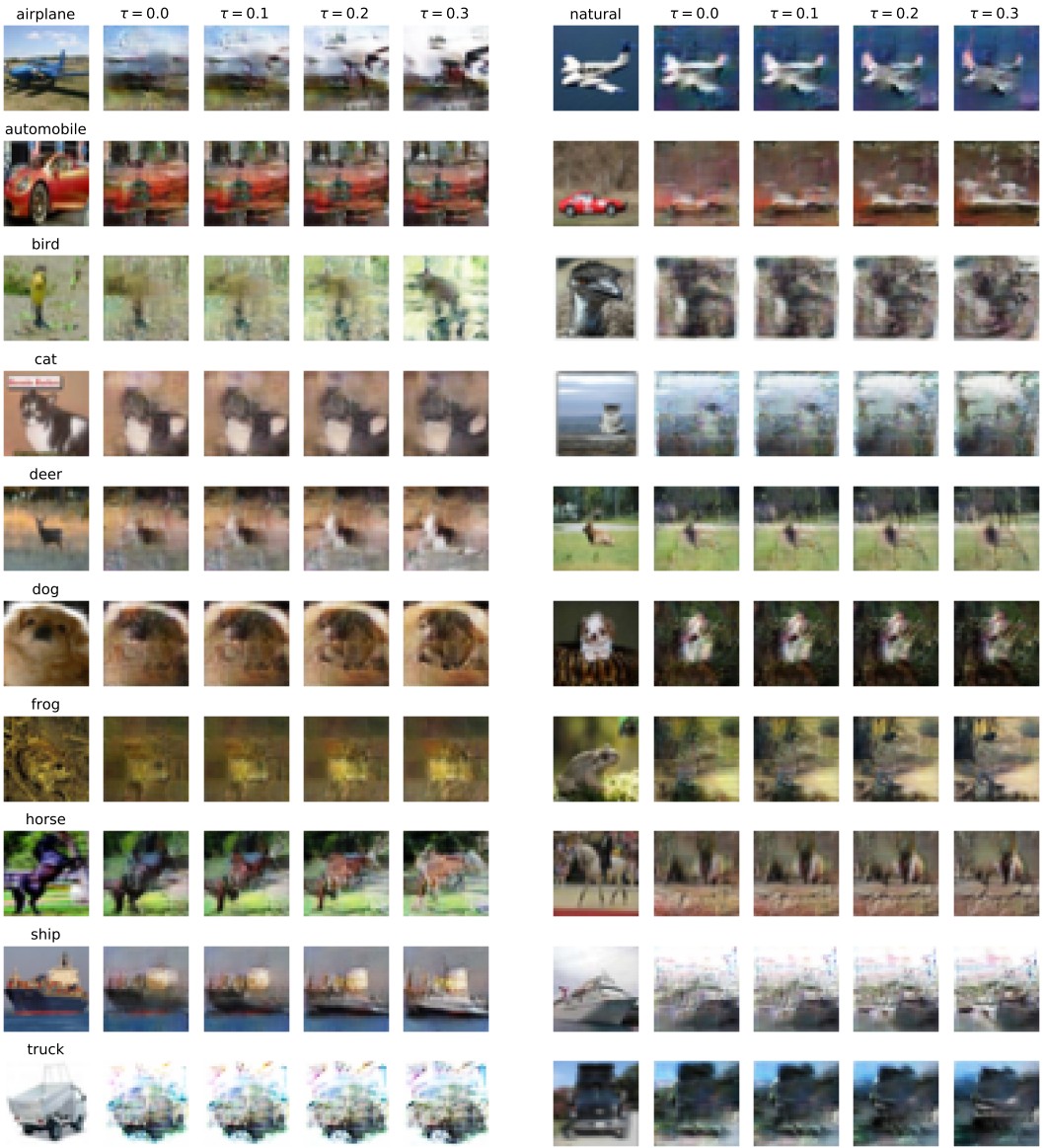

## C.3.2   $\ell_p$-BOUNDED ADVERSARIAL EXAMPLE ATTACKS

Each row shows two sets of randomly chosen examples from a single class. In each set, the leftmost image is the original image from the CIFAR-10 dataset and the subsequent images are the corresponding image perturbed using $\ell_p$-norm adversarial perturbations. At the top of the first row, each column's norm and $\varepsilon$ bound is given. For both the $\ell_2$ and $\ell_\infty$ norm-bounded examples, the highest tested $\varepsilon$ frequently perturb the image sufficiently to result in an apparent change of class. At the moderate $\varepsilon$, these class changes are rare. At the lowest tested $\varepsilon$, the images do not appear substantially different, even when comparing side-by-side.

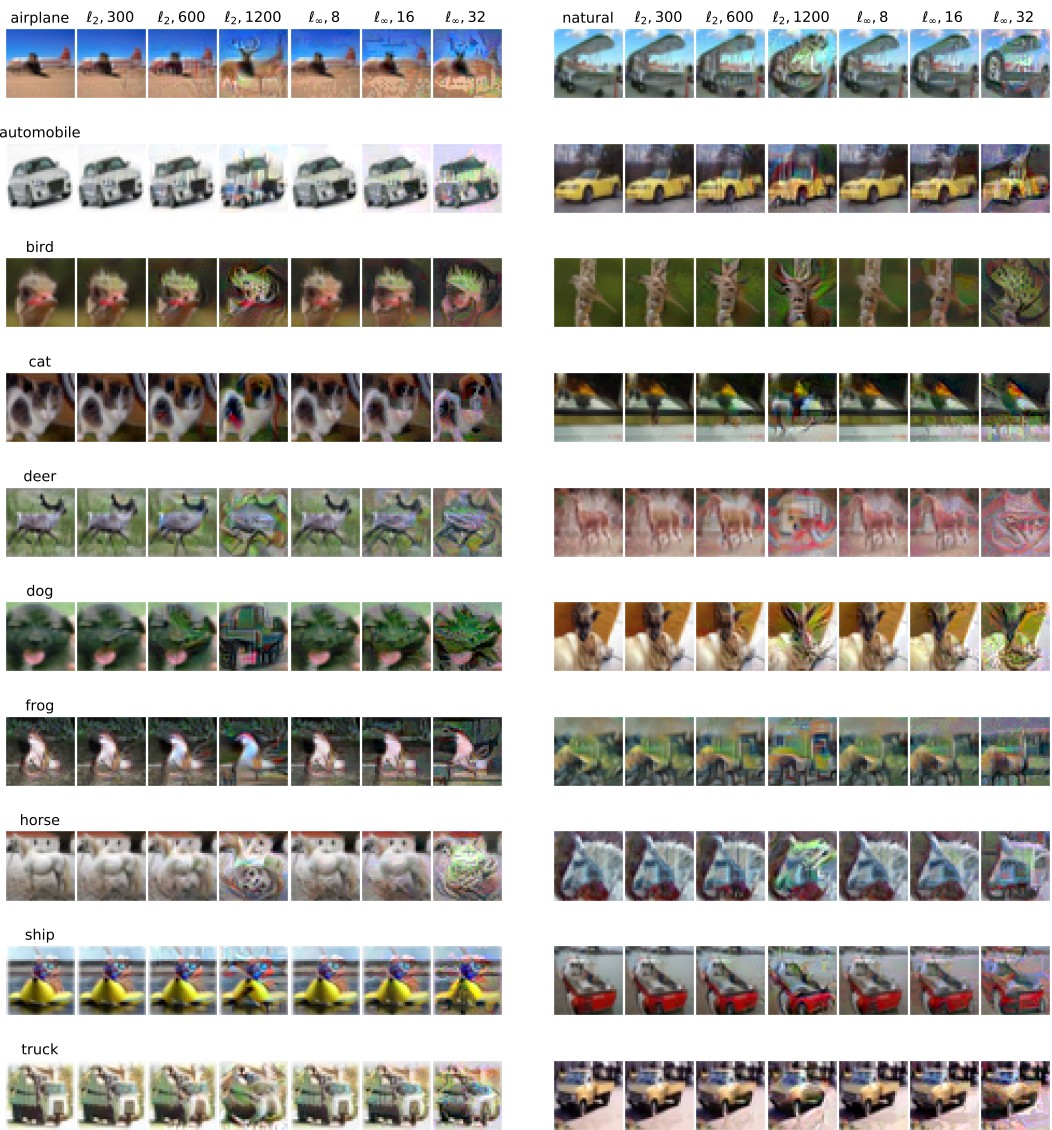

### C.3.3 Reduced amplitude attacks

Each row shows five pairs of examples from a single class. The left image in each pair is the original image from the CIFAR-10 dataset and the right image is the corresponding image perturbed using $\ell_2$-norm adversarial perturbations (bounded by $\varepsilon = 600$) and with the reduced amplitude backdoor pattern applied (using an amplitude of 16). It should be noted that the poisoned images do rarely change enough to appear mislabeled.

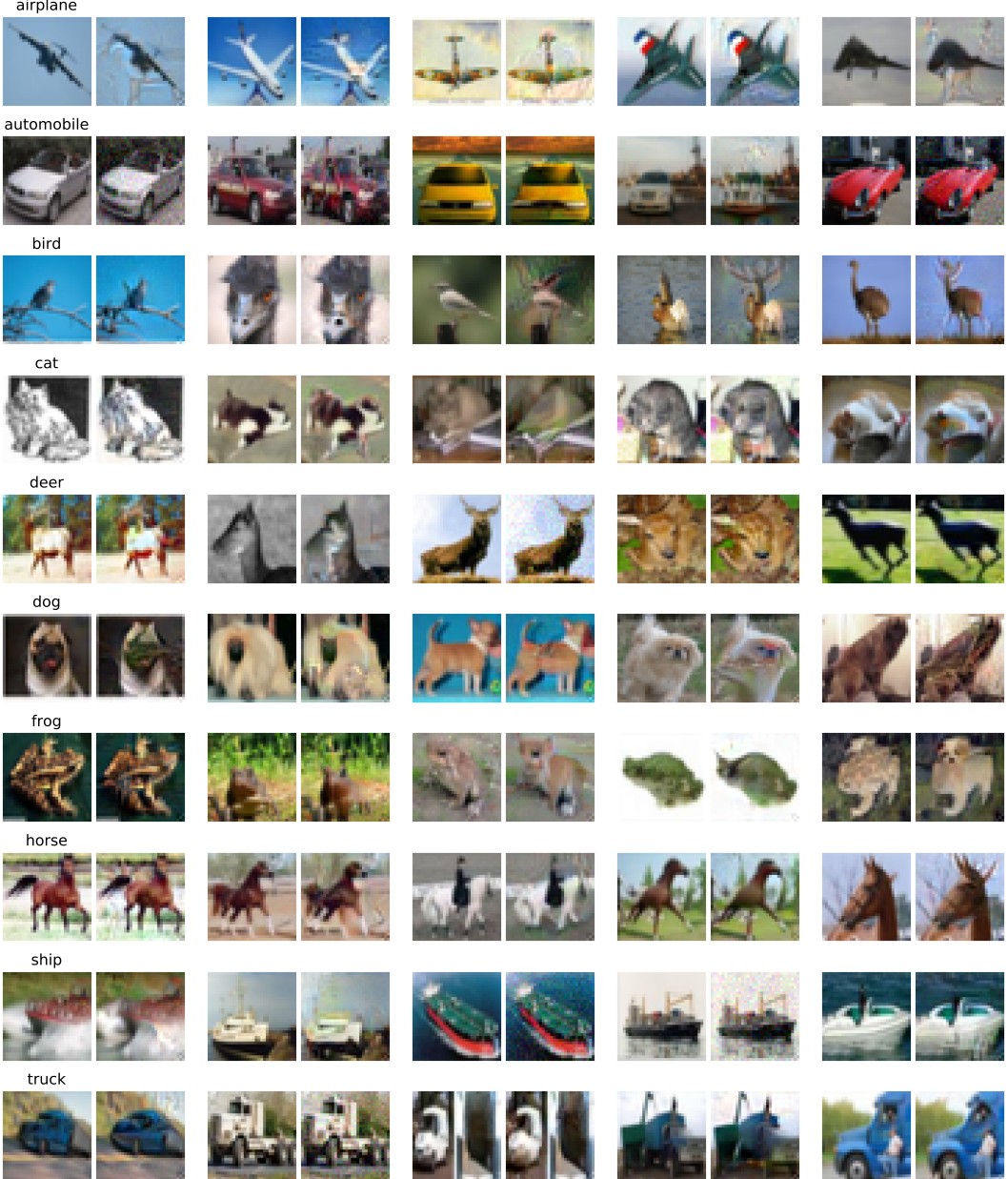

