# OpenReview forum: "Clean-Label Backdoor Attacks"
_ICLR.cc/2019/Conference_

### Official Review · AnonReviewer1 · 2018-10-31
**A nice idea which needs further in-depth exploitation**

**Rating:** 4
**Confidence:** 4

**Review:**

This paper investigates an interesting problem, backdoor attack against neural networks. The main idea is to add a watermark pattern to the corners of the training images, so that the classifier is guided to leverage the watermark as a discriminative cue as opposed to the real content of the image. At the test stage, one can hence manipulate the classifier’s predictions by adding the watermark to the test images.

This paper is heavily built upon Gu et al. (2017)’s work. It shows that Gu et al. (2017)’s method can be easily defended by a data sanitization algorithm. To improve Gu et al.’s work, the authors propose to add watermark patterns to the adversarial examples or examples interpolated in GAN’s latent space. The intuition is that these examples are adversarial and hard to learn, forcing the classifier to focus on the watermark pattern instead.

It is an interesting idea and an intuitive improvement over (Gu et al. 2017). However, the implementation of the idea could be improved. This paper does not propose any new attack algorithms. Instead, it investigates an existing adversarial attack method and the GAN based interpolation for the purpose of backdoor attack. As experiments are conducted on small-scale datasets, it is unclear how effective the improved backdoor attack is. Moreover, one of the main disadvantages of the proposed attack method is that simple data augmentation techniques, especially random cropping, can successfully defend against the attack.

The quality of the paper writing could be improved. I had to read the paper more than twice and check the references now and then in order to understand some claims of the paper. The paper’s lack of clarity was actually also raised by probably one of the coauthors of the paper; see the comment “Dimitris: clarify this point” on Page 11. Please find some concrete suggestions below.
- Figure 1 is visually not appealing at all. Perhaps find better illustrative examples.
- It is worth considering to add a separate section/paragraph to describe the details of Gu et al. (2017)’s method, given that this paper is heavily built upon Gu et al. (2017)’s work.
- It was unclear what the “reduced amplitude backdoor trigger” means until Section 4. If a context-dependent term has to be used in the introduction, explain it or refer the readers to the right place of the paper.
- Merge Sections 4.3—4.5 with the experiment section (Section 5). The results of Section 4.3–4.5 are out of context without any explanation about the experiment setups.

I have some concerns about Section 3, which is the main motivation of this work. As the authors noted in Appendix A that Gu et al.’s method works well with as few as 75 poised examples, the proposed sanitization algorithm would not be able to fail Gu et al.’s method by only identifying 20 out of 100 poised examples.

How to control the parameter $\tau$ so that the perturbation appears plausible to humans?

---

> ### Author Response · Authors · 2018-11-19
> **Author response**
>
> We thank the reviewer for the thoughtful comments. We will address comments raised below.
>
> - On the novelty of our attacks. We believe that the main conceptual contribution of our work is the formulation of the clean-label attack problem and showing how these attacks can be made successful by modifying samples to be "harder". The two attacks investigated are meant as proof-of-concept that this approach works with existing methods. We agree with the reviewer that designing specialized attacks for this task is a valuable research direction that could lead to even more successful attacks.
>
> - On the scale of our datasets. We do not have the resources to run an equally comprehensive study on ImageNet-scale datasets. Hence, we decided to perform more experiments on a small dataset rather than fewer experiments on a larger dataset. Note that the plots in Figure 3 and 4 involved training 50 models each. Does the reviewer have concrete concerns about the applicability of our approach to large-scale datasets?
>
> - On the resistance of our approach to data augmentation. We have demonstrated (Appendix B) that simply modifying the pattern to appear in all 4 corners is already sufficient to make the attack significantly more resistant to data augmentation. Thus, we don't consider data augmentation to be a fundamental obstacle to our attack. We believe that future work investigating different backdoor triggers can further increase the resistance of our attack to data augmentation.
>
> We thank the reviewer for concrete suggestions on improving our manuscript. We incorporated the following changes:
> - We replaced Figure 1 with more illustrative examples.
> - We modified the second paragraph of Section 2 to better explain the original Gu et al. (2017) attack.
> - We changed the wording of “reduced amplitude backdoor trigger” to "less conspicuous backdoor trigger" which should be clear without any further context.
> - The goal of Sections 4.3 - 4.5 is to provide a reader with an overview of our results before going into the experimental details. We modified these Sections to be more self-contained.
>
> - On concerns about Section 3. We do not argue that manual inspection will find all the poisoned examples (or enough to render the attack ineffective). We rather argue that if manual inspection of 300 images reveals 20 *clearly mislabelled* images, then the attack will very likely be detected leading to additional investigation and filtering. This argument illustrates a broader point -- if poisoned inputs appear suspicious upon human inspection, the attack is not truly insidious and can always be detected by more advanced filtering. This is why we believe our proposed attack is powerful: even if the samples are identified as potential outliers, they will not appear suspicious upon human inspection. We modified the text to better explain our argument.
>
> - We chose the parameter tau by manually inspecting different values of \tau on a 100 images.

---

> > ### Comment · AnonReviewer1 · 2018-11-27
> > **Appreciate the efforts on improving the manuscript; Concerns remain**
> >
> > Thank the authors for improving the manuscript and clarifying some details.
> >
> > Unfortunately, I still think the idea is interesting and yet the implementation of the idea is poor. This assessment is acknowledged by the authors by that the two attacks investigated are meant as proof-of-concept only. It is hard to extend the proof-of-concept methods to larger-scale datasets because the latent space of GAN does not naturally possess any disentangling properties, likely giving rise to unexpected output of the interpolation over the latent space. Besides, it is also hard to make the proof-of-concept methods robust against data augmentation --- Appendix B is unfortunately not convincing to me. Still, random cropping, mean subtraction, and rotation among other randomization techniques could increase a neural network's robustness to the changes at the image corners.

---

> > > ### Author Response · Authors · 2018-11-29
> > > **Author response**
> > >
> > > We are somewhat puzzled by the reviewer's criticism that our results are "proof-of-concept only".
> > >
> > > After all, our goal is to explore the space of possible attacks and understand their power. To this end, we have proposed a new approach to backdoor attacks (clean labels with hard samples) that addresses shortcomings of previous methods (implausible labels). We have evaluated our approach on a concrete, realistic learning problem with concrete attacks and showed that it can be effective.
> > >
> > > In the light of this goal, scaling the approach to larger datasets and making our attack robust against all possible defenses falls beyond the scope. One could argue that, from such a perspective, most research is indeed "proof-of-concept only".
> > >
> > > In regard to the reviewer's concerns that this approach would not scale to a larger scale dataset, we are not sure we understand the basis of that concern.
> > >
> > > Yes, it is possible that, for large scale datasets, the latent space of current GANs is not as well behaved. However, this is not a limitation of our method per se. After all, as GANs continue to become better, our approach will also improve with them, and research progress on GANs has been impressive over the last few years. As an example, we would like to point the reviewer to Figure 8 of a concurrent ICLR'19 submission (https://openreview.net/pdf?id=B1xsqj09Fm). That figure presents visually striking cross-class interpolations, which is all our attack requires.
> > >
> > > At the same time, our second approach, the perturbation-based attack, faces no such difficulties and can be applied as-is to large scale datasets (see, e.g., high-resolution images from Tsipras et al., 2018, https://arxiv.org/abs/1805.12152).
> > >
> > > Finally, the reviewer is concerned about the robustness of our approach to data augmentation. We want to emphasize that the networks of Appendix B were trained with random flips _and_ random crops. Moreover, we forgot to mention that those models also perform per image standardization (mean subtraction and standard deviation scaling). The results of Appendix B follow _exactly_ the input preprocessing pipeline used by state-of-the-art models without any modification on our part. (Random rotations are not part of any CIFAR-10 or ImageNet pipeline that we are aware of). Nevertheless, our attack remains successful in this regime. (More broadly, as data augmentation techniques tend to be standardized and well known, once the attacker knows what they are, they would be able to ensure the triggers 'survive' them.)

---

> > > > ### Comment · AnonReviewer1 · 2018-12-16
> > > > **The responses are unnecessarily defensive**
> > > >
> > > > It is unfortunate that the responses are unnecessarily defensive. Nonetheless, it seems that the authors have understood the basis of my concerns according to their responses. Hence, I will refrain from any further clarification.

---

### Official Review · AnonReviewer2 · 2018-10-31
**I think this paper adds an original and valuable angle to the existing literature on data poisoning attacks**

**Rating:** 7
**Confidence:** 2

**Review:**

Overall I am positive about this manuscript:
- I find the motivation is clear and valid. As far as I know, this is a novel contribution (my confidence is not very high on that one though - I might be unaware of related work).
- The paper is well-written and organized.
- Experiments are conducted systematically, although certain parts could be better explained (see my questions below).

I think this paper adds an original and valuable angle to the existing literature on data poisoning attacks. I don't see any major flaws, therefore I think it should be accepted.

A few points which might need clarification:
- How exactly is "attack success" being measured?
- Which model is used to generate the adversarial samples? Is this an (adversarially) pretrained model? (If that's the case, then what is the model architecture?) Or are adversarial samples generated on the fly using the currently trained/poisoned model?
- At the end of Section 4.4: if the images with larger noise rely more on the backdoor, why does this have an adverse effect? Shouldn't it increase the effectiveness of the attack?
- Was the data augmentation (flips, crops etc) performed before or after the poisoning pattern was applied?

Minor comments:
- definition of the encoding at the bottom of page 4: this should be argmax instead of max
- typo in Sec. 5.1: "to evaluate the uat a wide variety"
- repetitive sentence in Sec. 5.2: "we find that images generated with $\tau \leq 0.2$ remain [fairly] plausible"

---

> ### Author Response · Authors · 2018-11-19
> **Author response**
>
> We thank the reviewer for the kind comments and helpful suggestions. We will address points raised below:
> - The attack success rate (ASR) is computed as the fraction of inputs that are _not_ labeled with the target class but are classified as the target class after the backdoor pattern is applied (Beginning of Section 5). We have edited the manuscript to make this definition appear more prominently earlier in the paper and edited the relevant captions.
> - We use adversarially trained models trained with the publicly available code from https://github.com/MadryLab/cifar10_challenge (we train the non-wide variant both with L2 and Linf). The adversarial examples are generated once using this pre-trained network. Since our threat model only allows us to add examples to the training set, we cannot compute these adversarial perturbations on the fly. We have edited the manuscript to incorporate this discussion.
> - We were also surprised initially but we believe that there is a fairly simple explanation (outlined in Section 4.4). On noisy images, the classifier learns to predict by relying on the backdoor *in the absence of strong image signal* (since the salient image features are fairly corrupted). However, when evaluated on the test set with a backdoor applied, the image itself will have a strong signal (since it will not be noisy) that can overcome the backdoor pattern. Therefore, it is necessary for the classifier to learn to predict the backdoor even when the salient image characteristics are present. As a result, random noise is not very effective at injecting backdoors. We have updated Section 4.4 to better reflect this argument.
> - Since we do not have access to the training procedure, the pattern is applied before any data augmentation. This is the reason why this setting is challenging -- data augmentation might obscure the pattern.
>
> We have updated the manuscript to incorporate the other comments.

---

> > ### Comment · AnonReviewer2 · 2018-11-25
> > **Thank you for the clarifications!**
> >
> > Those answer all the questions I had.

---

> > > ### Comment · AnonReviewer1 · 2018-12-16
> > > **Check the other reviews in addition to the authors' responses to your comments?**
> > >
> > > Dear AnonReviewer2,
> > >
> > > I am writing to bring your attention to my comments below, per the Area Chair's request. In my opinion, this paper's overarching idea is very interesting and yet the implementation could be improved. In particular, I had three major concerns in my initial review.
> > >
> > > 1. This paper does not propose any new attack algorithms. Instead, it investigates an existing adversarial attack method and the GAN based interpolation for the backdoor attack.
> > > 2. As experiments are conducted on small-scale datasets, it is unclear how effective the improved backdoor attack is.
> > > 3. Moreover, one of the main disadvantages of the proposed attack method is that simple data augmentation techniques, especially random cropping, can successfully defend against the attack.
> > >
> > > The first two concerns were reinforced by the authors' responses (at least from my point of view). The answer to the third concern is not convincing. My background is largely from computer vision, and I can think of many data augmentations to overcome the four-corner backdoor patterns studied in this paper.
> > >
> > >
> > > Best regards,
> > > AnonReviewer1

---

### Official Review · AnonReviewer3 · 2018-11-07
**Clean-Label Backdoor Attacks**

**Rating:** 6
**Confidence:** 2

**Review:**


This work explores backdoor attacks -- attacks that alter a fraction of training examples which can alter inference -- while ensuring that the poisoned inputs are consisten
t with their labels. These attacks are attained through either a GAN mechanism or using adversarial perturbations.

The ideas proposed (i.e. GAN mechanism and adversarial mechanism) are interesting additions to this literaature. I found the observation of greater effectiveness of adversa
rial mechanism particularly interesting.

The paper also does a good job of investigating effectiveness of the attack under data augmentation and propooses a limited solution.

Main criticism: there are a number of typos that need fixing.
~

---

> ### Author Response · Authors · 2018-11-19
> **Author response**
>
> We thank the reviewer for the kind comments. We have updated the manuscript to fix typos.

---

### Public Comment · ~Mohammad_Mahmoody1 · 2018-11-27
**Earlier attacks on classifiers using clean/correct labels**

This interesting work considers backdoor attacks that involve poisoning using correctly labeled examples. This is indeed a very interesting direction. As the focus of the paper is to show the power of correctly-labeled poisoned data in attacks on classifiers, naturally the paper cites a recent work on targeted poisoning attacks using correctly labeled examples.

Here we would like to mention some earlier works (some from 2017) showing the power of (targeted) poisoning attacks using *correctly-labeled* examples in the attack. We hope these references will be found useful.

All of the previous attacks listed below *provably* apply to *any* classification task and *any* classifier (so obviously they apply to neural networks as well).

1. In a TCC 2017 paper titled “Blockwise p-Tampering Attacks on Cryptographic Primitives, Extractors, and Learners”  (online since Sept 2017)
https://eprint.iacr.org/2017/950
it was shown that by changing p fraction of the training examples (and planting still correctly labeled examples instead of them) the adversary can increase classification error of any particular instance x to go up by at least Omega(p) (starting from any initial small constant error like 0.001 over the learning and testing phases).

2. In an “Algorithmic Learning Theory (ALT) 2018” paper titled “Learning under p-Tampering Attacks” (online since Nov 2017)
https://arxiv.org/abs/1711.03707
the constants in the bounds of the TCC-17 paper were improved.

In the above two works (1,2), attacks using correct labels are referred to as “defensible” attacks (inspired by similar terms used in Cryptography) as they could be “defended” due to the use of correct labels.

3. In an AAAI 2019 paper “The Curse of Concentration in Robust Learning...” (online since 9 Sept 2018)
https://arxiv.org/abs/1809.03063
it was shown that the adversary can do the same job by substituting way fewer examples (namely sqrt(m) ones, where m is the sample complexity) with other correctly labeled examples.

4. The first 2 attacks above are polynomial-time, while the 3rd one is existential. In a more recent work https://arxiv.org/abs/1810.01407 (online since Oct 2)
it is shown how to get the best of the worlds above; namely in order to increase the targeted error from an arbitrary small constant to an arbitrary large constant, a polynomial-time adversary only needs to substitute O(sqrt (m)) of the examples (where m is the sample complexity) with other still correctly labeled ones.

The latter two papers (3,4) above, refer to attacks using correct labels as “plausible” attacks (which seems to be the term also used in this paper).

All the attacks above are stated in the targeted poisoning scenario with the goal of increasing the classification *error* of a target instance (e.g., the true label is 0, while the adversary wants to get a label other than 0). However the same proofs (as is) apply even if the attacker wants to make the target instance x get a specific label \ell with a probability close to 1 assuming that originally (without any attacks) the probability of x being labeled \ell (over the whole training and testing processes) is at least an arbitrary small constant.

---

> ### Author Response · Authors · 2018-11-29
> **Thank you for the suggestions**
>
> We thank Mohammad for bringing this line of work to our attention. This is a very interesting theoretical study of the phenomenon that we will make sure to cite and discuss in future versions of our manuscript.
>
> For the sake of the discussion, we would like to note though that the attacks explored here are _targeted_ poisoning attacks and not _backdoor_ attacks such as those considered in our paper.

---

> > ### Public Comment · ~Mohammad_Mahmoody1 · 2018-11-30
> > **Yes! previous comment was about the type of labels used**
> >
> > Indeed, backdoor attacks (studied here) are different from targeted (and other types of) poisoning attacks (studied in papers I listed), though they both have a poisoning phase that might or might not use (only) correct/clean labels.
> >
> > My comment was only meant to point out the earlier works that also use (only) clean labels in the poisoning phase of the attack and hope that you find them useful!

---

### Meta-Review · Area_Chair1 · 2018-12-18
**interesting idea, good execution, but just below threshold**

**Confidence:** 2
**Recommendation:** Reject

**Metareview:**

The present work proposes to improve backdoor poisoning attacks by only using "clean-label" images (images whose label would be judged correct by a human), with the motivation that this would make them harder to detect. It considers two approaches to this, one based on GANs and one based on adversarial examples, and shows that the latter works better (and is in general quite effective). It also identifies an interesting phenomenon---that simply using existing back-door attacks with clean labels is substantially less effective than with incorrect labels, because the network does not need to modify itself to accommodate these additional correctly-labeled examples.

The strengths of this paper are that it has a detailed empirical evaluation with multiple interesting insights (described above). It also considers efficacy against some basic defense measures based on random pre-processing.

A weakness of the paper is that the justification for clean-label attacks is somewhat heuristic, based on the claim that dirty-label attacks can be recognized by hand. There is additional justification that dirty labels tend to be correlated with low confidence, but this correlation (as shown in Figure 2) is actually quite weak. On the other hand, natural defense strategies against the adversarial examples based attack (such as detecting and removing points with large loss at intermediate stages of training) are not considered. This might be fine, as we often assume that the attacker can react to the defender, but it is unclear why we should reject dirty-label attacks on the basis that they can be recognized by one detection mechanism but not give the defender the benefit of other simple detection mechanisms for clean-label attacks.

A separate concern was brought up that the attack is too similar to that of Guo et al., and that the method was not run on large-scale datasets. The Guo et al. paper does somewhat diminish the novelty of the present work, but not in a way that I consider problematic; there are definitely new results in this paper, especially the interesting empirical finding that the Guo et al. attack crucially relies on dirty labels. I do not agree with the criticism about large-scale datasets; in general, not all authors have the resources to test on ImageNet, and it is not clear why this should be required unless there is a specific hypothesis that running on ImageNet would test. It is true that the GAN-based method might work more poorly on ImageNet than on CIFAR, but the adversarial attack method (which is in any case the stronger method) seems unlikely to run into scaling issues.

Overall, this paper is right on the borderline of acceptance. There are interesting results, and none of the weaknesses are critical. It was unfortunately the case that there wasn't room in the program this year, so the paper was ultimately rejected. However, I think this could be a strong piece of work (and a clear accept) with some additional development. Here are some ideas that might help:

(1) Further investigate the phenomenon that adding data points that are too easy to fit do not succeed in data poisoning. This is a fairly interesting point but is not emphasized in the paper.
(2) Investigate natural defense mechanisms in the clean-label setting (such as filtering by loss or other such strategies). I do not think it is crucial that the clean-label attack bypasses every simple defense, but considering such defenses can provide more insight into how the attack works--e.g., does it in fact lead to substantially higher loss during training? And if so, at what stage does this occur? If not, how does it succeed in altering the model without inducing high loss?